# The response of soil Olsen-P to the P budgets of three typical cropland soil types under long-term fertilization

**Weiwei Zhang**[1], **Qiong Wang**[1¤], **Qihua Wu**[2], **Shuxiang Zhang**[1]*, **Ping Zhu**[3], **Chang Peng**[3], **Shaomin Huang**[4], **Boren Wang**[1], **Huimin Zhang**[1]

**1** Institute of Agricultural Resources and Regional Planning, Chinese Academy of Agricultural Sciences/ National Engineering Laboratory for Improving Quality of Arable Land, Beijing, P. R. China, **2** Guangdong Provincial Bioengineering Institute (Guangzhou Sugarcane Industry Research Institute)/Guangdong Key Laboratory of Sugarcane Improvement and Biorefinery, Guangzhou, P. R. China, **3** Centre of Agricultural Environment and Resources, Jilin Academy of Agricultural Sciences, Changchun, P. R. China, **4** Institute of Plant Nutrition, Resources and Environment, Henan Academy of Agricultural Sciences, Zhengzhou, P. R. China

¤ Current address: Institute of Agricultural Resources and Regional Planning, Chinese Academy of Agricultural Sciences, Beijing, P. R. China
* zhangshuxiang@caas.cn

**Data Availability Statement:** All relevant data are within the paper and its Supporting Information files.

## Abstract

The Olsen phosphorus (Olsen-P) concentration of soil is generally a good indicator for estimating the bioavailability of P and environmental risk in soils. To maintain soil Olsen-P at adequate levels for crop growth and environmental sustainability, the relationship between soil Olsen-P and the P budget (the P input minus the output) as well as the variations of soil Olsen-P and P budget were investigated from three long-term (22 years) experiments in China. Five treatments were selected: (1) unfertilized control (CK); (2) nitrogen and potassium (NK); (3) nitrogen, phosphorous, and potassium (NPK); (4) nitrogen, phosphorous, potassium and straw; (5) nitrogen, phosphorous, potassium and manure. The results showed that without P fertilizers (CK, NK), there was a soil P deficit of 75–640 kg ha$^{-1}$, and the lowest P deficit (mean of CK and NK) was in Eutric Cambisol. Soil Olsen-P decreased by 0.11–0.39 mg kg$^{-1}$ year$^{-1}$ in the order of Luvic Phaeozems > Eutric Cambisol > Calcaric Cambisol. Soil Olsen-P and the P deficit had a significantly ($P<0.01$) positive linear relationship. For every 100 kg of P ha$^{-1}$ of deficit, soil Olsen-P decreased by 0.44–9.19 mg kg$^{-1}$ in the order of Eutric Cambisol > Luvic Phaeozems > Calcaric Cambisol. Under the P fertilizer treatments (NPK, NPKS, and NPKM), soil Olsen-P showed an obvious surplus (except the NPK and NPKS in Luvic Phaeozems) of 122–2190 kg ha$^{-1}$, and the largest P surplus was found under the NPKM treatment at each site. The relation between soil Olsen-P and the experimental years could be simulated using quadratic equation of one unknown in Calcaric Cambisol for the lower P input after 14 years of fertilization. And soil Olsen-P increased by 1.30–7.69 mg kg$^{-1}$ year$^{-1}$ in the order of Luvic Phaeozems > Eutric Cambisol. The relation between soil Olsen-P and the P surplus could be simulated by a simple linear equation except under NPK and NPKS in Luvic Phaeozems. With 100 kg ha$^{-1}$ P surplus, soil Olsen-P increased by 3.24–7.27 mg kg$^{-1}$ in the order of Calcaric Cambisol (6.42 mg kg$^{-1}$) > Eutric Cambisol (3.24 mg kg$^{-1}$). In addition, the change in soil Olsen-P with a 100 kg P ha$^{-1}$ surplus (soil Olsen-P efficiency) was

**Funding:** The National Key Research and Development Program of China (2016YFD0300803); The National Natural Science Foundation of China (41471249); The Special Fund for Agro-scientific Research in the Public Interest of China (201503120). The funder was the corresponding author, who was responsible for data collection, decision to publish and correct the manuscript.

**Competing interests:** The authors have declared that no competing interests exist.

affected by the soil organic matter (SOM), pH, and $CaCO_3$ content, etc. In the practice of fertilization, it's not necessary to increase the amount of P fertilizers, farmers should take measure to solve the local problem, for adjust the soil pH of Eutric Cambisol and Calcaric Cambisol, and apply more nitrogen in Luvic Phaeozems. In the area of serious soil P surplus, it is encouraged to stop applying P fertilizers for a few years to take advantage of soil accumulated P and make the high Olsen-P content decrease to a reasonable level.

## Introduction

Phosphorus (P) is an essential element for plant growth. In an agricultural ecosystem, P fertilization is the most common practice for guaranteeing the crop yield [1, 2]. A large amount of residual P was accumulated in cultivated soils after long-term P overfertilization [3, 4, 5], and it was converted into less soluble and more stable forms, resulting in a low P use efficiency (PUE) (10–50%) [6, 7]. Excessive P fertilizers caused soil Olsen-P to rapidly increase and resulted in a risk of nonpoint source pollution [8, 9], but the crop yields were not improved by much [10]. Therefore, it is necessary to investigate the dynamic characteristics of soil Olsen-P to improve the PUE and reduce environmental pollution.

Many studies have focused on the soil Olsen-P response to different P fertilizer applications [11, 12, 13]. However, during long periods, the change in soil Olsen-P was primarily driven by the P budget due to many years of P removal from crop harvests and P fertilizer inputs [14, 15, 16]. Many long-term field experiments have established a significantly positive linear correlation between soil Olsen-P and the P budget, and with a 100 kg ha$^{-1}$ P surplus, soil Olsen-P increased by different rates in different soil types [4, 17, 18]. For black loess soils in Gansu, China, the Olsen-P concentration decreased by 3.18 mg kg$^{-1}$ (control) and 1.95 mg kg$^{-1}$ (nitrogen fertilizer only) per 100 kg ha$^{-1}$ of P deficit and increased by 0.29–3.85 mg kg$^{-1}$ per 100 kg ha$^{-1}$ of P accumulation when P (chemical P and manure) was added [19]. In black soil in Haerbin, China, the Olsen-P decreased by 3.35, 2.43 and 1.39 mg kg$^{-1}$ for the CK, N and NK treatments and increased by 4.8, 7.75 and 6.95 mg kg$^{-1}$ for the NP, NPK and NPKM treatments [18]. The variations in the soil Olsen-P response to the budget might be attributed to the different environments, crop systems, P inputs and soil properties, such as the soil organic matter and pH [4, 18]. Thus, understanding the variations and the possible factors affecting the relationship between the Olsen-P and the P budget is useful for predicting the Olsen-P dynamics and the optimal P fertilization of different soil types.

Luvic Phaeozems, Calcaric Cambisol, and Eutric Cambisol are the three soil types used in this study, and they arise from the northeastern, central and southern parts of China, where the primary agricultural regions are located. Five treatments were selected for this 22-year (1990–2012) long-term fertilization experiment. We addressed the effects of no P fertilizers and different P fertilizers on (1) the soil Olsen-P content; (2) the P budget; (3) the relationships between soil Olsen-P and the P budget; (4) the possible influence factors (soil organic matter, pH) to provide reasonable suggestions for the persistent and efficient utilization of P resources on the different soil types of China.

## Materials and methods

### Experimental sites

The three long-term experiment sites, which were established in 1990, are located in Gongzhuling (GZL), Jilin province, Northeast China; Zhengzhou (ZZ), Henan province, Central

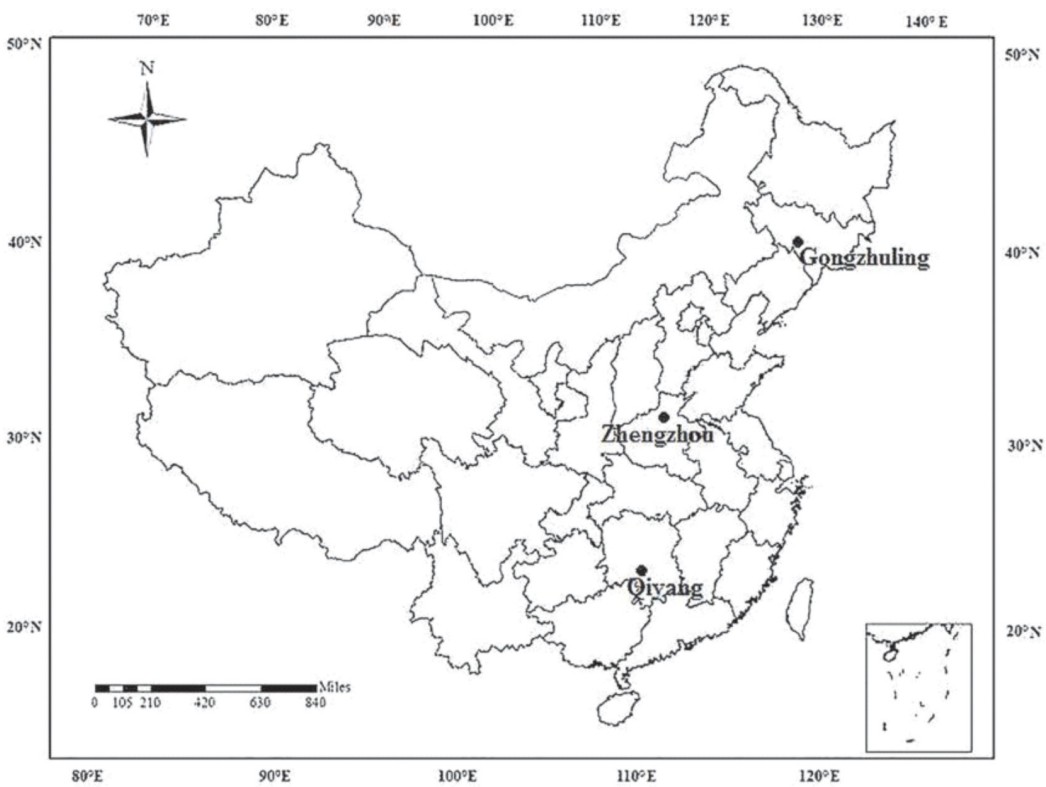

**Fig 1. Locations of the three long-term fertilizer application sites in China.**

China; and Qiyang (QY), Hunan province, South China (Fig 1) [20]. The study period of the three sites is from 1990 to 2012. The soils at the three sites are classified as Luvic Phaeozems in GZL, Calcaric Cambisol in ZZ, and Eutric Cambisol in QY [FAO]. The description of the three sites and the initial physicochemical properties of the surficial soil (0–20 cm) in 1990 are summarized in Table 1 [20].

## Experimental design

The cropping systems were mono-maize cropping at GZL (late April to late September), wheat-maize double-cropping at ZZ (mid-October to early June for wheat and mid-June to late September for maize) and QY(early November to early May for wheat and early April between wheat strips to July for maize). Five treatments (in a randomized plot) were selected in each site: (1) unfertilized control (CK); (2) nitrogen and potassium (NK); (3) nitrogen, phosphorous, and potassium (NPK); (4) nitrogen, phosphorous, potassium and straw (NPKS); (5) nitrogen, phosphorous, potassium and manure (NPKM). The plot area in GZL and QY was 200 $m^2$, and in ZZ was 45 $m^2$. There were no replications in the three sites. In order to analyze the spatial variation of soil and plant indicators, each individual treatment plot was divided into three subplots. Urea, superphosphate and potassium chloride were used as N, P, K fertilizers, respectively at the three sites. The straw was maize straw. The manure was pig manure in GZL and QY, and horse and cattle manure in ZZ. Chemical fertilizers were applied and ploughed into the soil one time before the plots were seeded with wheat and maize each year, and organic fertilizers (straw and manure) were applied prior to seeding plots with winter wheat each year in ZZ and QY. In GZL, the inorganic and organic fertilizers were applied

**Table 1. Locations, climate conditions (shown as the mean for 1990–2012, as obtained from the China meteorological sharing service system (http://cdc.cma.gov. cn/)), and the initial surficial soil properties (1990) of the three sites.**

| Parameters | Gongzhuling | Zhengzhou | Qiyang |
|---|---|---|---|
| Altitude (m) | 220 | 21 | 120 |
| Latitude (N) | 43˚30′ | 34˚47′ | 26˚45′ |
| Longitude (E) | 124˚48′ | 113˚40′ | 111˚52′ |
| Mean annual precipitation (mm) | 590.7 | 641 | 1426.4 |
| Mean annual temperature (˚C) | 6.6 | 14.7 | 18.0 |
| Cumulative effective temperature (>10˚C) | 2800 | 5169 | 5600 |
| Cropping system | Single-cropping, maize | Double-cropping, maize/wheat | Double-cropping, maize/wheat |
| Soil texture | Clay loam | Light loam | Light loam |
| Bulk density (g cm$^{-3}$) | 1.2 | 1.5 | 1.2 |
| Clay content (<0.002 mm, %) | 32.1 | 13.4 | 41.0 |
| Soil pH (soil: water = 1:2.5) | 7.6 | 8.3 | 5.7 |
| Organic matter (g kg$^{-1}$) | 22.8 | 11.6 | 13.6 |
| Total P (g kg$^{-1}$) | 0.6 | 0.6 | 0.5 |
| Olsen P (mg kg$^{-1}$) | 11.8 | 6.5 | 4.7 |
| CaCO$_3$ (g kg$^{-1}$) | 36.5 | 72.8 | 12.4 |
| Fe$_2$O$_3$ (g kg$^{-1}$) | 1.7 | 0.8 | 3.2 |
| Al$_2$O$_3$ (g kg$^{-1}$) | 1.4 | 0.6 | 2.1 |

annually before the seeding of maize. The average annual fertilizer application amounts are summarized in Table 2 [20].

## Soil and plant sampling and chemical analyses

The initial soil samples, before treatment application, were collected randomly from the arable layer (0–20 cm) from each site in 1990, because the soil total P pool was mainly within this layer of the three sites [21]. The soil samples from each treatment (plot) were collected annually from 1990 to 2012 after maize harvesting but before fertilizer application to each site. Soils from three subplots were treated as replications. An auger (5 cm internal diameter) was used to collect the soil samples from each plot. Three soil cores were collected from each subplot and were combined to form each composite sample. Thus, there were three soil samples and 9 soil cores in each plot. Each soil sample was subsequently air-dried and sieved through 2.0 mm mesh screens to determine the available nutrients, and then through 0.25 mm mesh screens prior to the total nutrient analyses. Soil Olsen-P was determined using 0.5 mol L$^{-1}$ sodium bicarbonate extraction (pH 8.5) and measured via molybdenum blue colorimetry method. The soil organic matter was measured using the oil bath-potassium permanganate volume method. The soil pH was also measured (mass/volume ratio of 1:2.5). The grains and straws were harvested manually, air-dried, threshed, oven-dried at 65˚C to a uniform moisture level, and then weighed. The P concentrations of the grain and straw were also measured using the molybdenum blue colorimetric method. The indices were analyzed in accordance with Lu [22].

## Calculation and statistical analysis

The P budget (kg P ha$^{-1}$) was calculated as the Σ [apparent P budget of crops in season].

The apparent soil P budget (kg P ha$^{-1}$) was calculated as the total amount of annual P fertilizers (kg P ha$^{-1}$)–the annual crop P uptake (grain + straw) (kg P ha$^{-1}$).

In the present study, the apparent P budget was equal to the crop uptake of P each year, which was the sum of the maize P uptake and wheat P uptake; the crop P uptake (kg ha$^{-1}$) was

**Table 2. Rates of N, P, and K application in the forms of chemical fertilizer and manure at the three long-term fertilizer application sites.**

| Treatments[a] | Gongzhuling | | Zhengzhou | | Qiyang | |
|---|---|---|---|---|---|---|
| | inorganic[b] N-P-K (kg ha⁻¹) | organic P (kg ha⁻¹) | inorganic N-P-K (kg ha⁻¹) | organic P (kg ha⁻¹) | inorganic N-P-K (kg ha⁻¹) | organic P (kg ha⁻¹) |
| CK | 0-0-0 | 0 | 0-0-0 | 0 | 0-0-0 | 0 |
| NK | 165-0-68 | 0 | 353-0-146 | 0 | 300-0-100 | 0 |
| NPK | 165-36-68 | 0 | 353-62-146 | 0 | 300-53-100 | 0 |
| NPKM[c] | 165-36-68 | 40.5 | 353-62-146 | 40 | 300-53-100 | 84 |
| NPKS[d] | 165-36-68 | 5.3 | 353-62-146 | 9.3 | 300-53-100 | 2.8 |

a CK: unfertilized control; NK: nitrogen and potassium; NPK: nitrogen, phosphorous, and potassium; NPKS: nitrogen, phosphorous, potassium and straw; and NPKM: nitrogen, phosphorous, potassium and manure (NPKM).

b Inorganic N fertilizer as urea, P as calcium triple superphosphate, and K as potassium sulfate. In ZZ, the $P_2O_5$ content of the calcium triple superphosphate decreased from 12.05% (1990–2003) to 8.0% (2004–2012).

c The manures were pig manure from 1990 at GZL (23.0 mg ha⁻¹ year⁻¹) and QY (42.0 mg ha⁻¹ year⁻¹), but horse manure was used from 1990 to 1998 and cattle manure was used from 1999 to 2012 at ZZ(12.9 mg ha⁻¹ year⁻¹). All the manure amounts were averaged as fresh weights from 1990 to 2012.

d The entire quantity of maize straw was incorporated into the soil at GZL (approximately 7.5 mg ha⁻¹) and ZZ (on average 6.0 mg ha⁻¹), whereas at QY, half of the maize and wheat straw (approximately 4.5 mg ha⁻¹) was applied.

calculated as the (grain yield (kg ha⁻¹) × grain P content (%)) + (straw yield (kg ha⁻¹) × straw P content (%)).

A simple linear model was used to determine the relationships between soil Olsen-P and the experimental years, soil Olsen-P and the P budget, soil Olsen-P efficiency and organic matter, soil Olsen-P efficiency and pH; quadratic equation of one unknown was used to determine the relationships between soil Olsen-P and the experimental years in Calcaric Cambisolin in Excel 2013 (Microsoft Corp, Redmond, Washington, USA). The SPSS 20.0 (International Business Machines Corporation, Armonk, NewYork, USA) was used to do the data analysis by calculating the mean, the standard error of mean of crop yield, soil pH and SOM. Before doing the ANOVA significance levels in Table 3, descriptive statistics was used in SPSS to ensure the normal distribution of data.

## Results

### Soil P budget

The P budget exhibited two types of trends in the soil P pool over time, which were a "deficit" and a "surplus" of soil P depending on the treatments (Fig 2). Under the treatments without P fertilizers (CK and NK), the soil P budget was negative and decreased over the experimental years. The P deficit ranged from -640 ~ -5 kg ha⁻¹ at the three sites and was ordered Luvic Phaeozems (-640 ~ -14 kg ha-1) < Calcaric Cambisol (-447 ~ -27 kg ha⁻¹) < Eutric Cambisol (-103 ~ -5 kg ha⁻¹). The P deficit was almost close in the treatments without P applications (CK and NK) in Eutric Cambisol and Calcaric Cambisol. However, the P deficit under NK (-640 ~ -28kg ha⁻¹) was significant ($P<0.05$) lower than that under CK (-290 ~ -14 kg ha⁻¹) in Luvic Phaeozems [Table 3].

Under the P fertilizer treatments (NPK, NPKS, and NPKM), the soil P showed a surplus of -12 ~ 2190 kg ha⁻¹ at the three site, and the P surplus under NPKM was significant ($P<0.05$) higher than that under NPK and NPKS in the three soil types [Table 3]. There was not much difference in the P surplus under NPK and NPKS in Eutric Cambisol and Calcaric Cambisol. In Calcaric Cambisol, after 14 years of cultivation, the soil P surplus decreased under NPK and NPKS and did not increase much under NPKM. In Luvic Phaeozems, the P surplus under NPKS was significantly higher than that under NPK ($P< 0.05$), the P surpluses under NPK and NPKS were relatively low, with ranges of -12~20 and 12~48 kg ha⁻¹ over the 22 years of cultivation.

**Table 3. Soil Olsen-P, P budget, crop yeil and soil properties (pH and SOM) under the three soil types and five treatments.** ANOVA significance levels for the effects of soil type, fertilization, and soil type and fertilization interactions.

| Treatments | P budget | Olsen-P | Maize yeild | Wheat yeild | SOM | pH |
|---|---|---|---|---|---|---|
| | kg ha⁻¹ | mg kg⁻¹ | t ha⁻¹ | t ha⁻¹ | g kg⁻¹ | |
| Eutric Cambisol | | | | | | |
| CK | -45.11(4.53)C | 5.50(0.60)C | 0.30(0.04)D | 0.36(0.03)C | 14.36(0.30)C | 5.71(0.07)A |
| NK | -86.68(5.42)C | 5.34(0.62)C | 1.18(0.29)C | 0.47(0.13)C | 14.84(0.37)C | 4.68(0.10)B |
| NPK | 385.52(51.86)B | 31.53(2.34)B | 2.88(0.33)B | 1.04(0.13)B | 16.77(0.38)B | 4.70(0.10)B |
| NPKS | 359.80(49.73)B | 35.05(2.29)B | 3.44(0.34)B | 1.14(0.12)B | 16.75(0.44)B | 4.73(0.10)B |
| NPKM | 1156.06(137.24)A | 85.31(11.22)A | 4.93(0.19)A | 1.73(0.09)A | 21.24(0.86)A | 5.91(0.07)A |
| Calcaric Cambisol | | | | | | |
| CK | -226.89(24.85)C | 3.28(0.25)C | 3.02(0.21)C | 1.75(0.09)C | 10.93(0.18)C | 8.45(0.05)A |
| NK | -242.36(25.56)C | 2.81(0.29)C | 4.07(0.25)B | 2.91(0.20)B | 11.96(0.15)A | 8.35(0.03)AB |
| NPK | 186.02(17.31)B | 17.52(1.77)B | 6.82(0.39)A | 6.53(0.22)A | 12.30(0.29)A | 8.35(0.03)AB |
| NPKS | 224.63(20.41)B | 18.45(1.72)B | 7.43(0.36)A | 6.42(0.21)A | 14.51(0.34)A | 8.33(0.04)B |
| NPKM | 607.86(62.97)A | 39.24(3.92)A | 7.17(0.38)A | 6.18(0.22)A | 15.37(0.49)A | 8.40(0.04)AB |
| Luvic Phaeozems | | | | | | |
| CK | -153.52(16.82)D | 5.38(0.61)C | 3.52(0.23)C | - | 14.88(0.78)C | 7.60(0.05)A |
| NK | -342.34(39.71)C | 6.37(0.84)C | 7.95(0.31)B | - | 15.85(0.77)BC | 6.70(0.12)C |
| NPK | 7.79(1.99)B | 19.34(2.22)BC | 9.02(0.37)A | - | 15.67(0.82)BC | 6.58(0.12)C |
| NPKS | 29.06(2.62)B | 29.15(2.96)B | 9.31(0.28)A | - | 18.21(0.86)B | 7.00(0.06)B |
| NPKM | 348.76(36.52)A | 77.56(14.14)A | 9.16(0.39)A | - | 25.70(1.33)A | 7.51(0.03)A |
| ANOVA | | | | | | |
| Soil type | ** | ** | ** | ** | ** | ** |
| Fertilization | ** | ** | ** | ** | ** | ** |
| Soil type×Ferlization | ns | ** | * | ** | ns | ** |

## Soil Olsen-P

Under the five treatments in each site, the order of soil Olsen-P content was NPKM > NPK and NPKS > CK and NK, and the difference of the three group was significant ($P<0.05$) (Table 3). The changes in soil Olsen-P were different under the various fertilization treatments

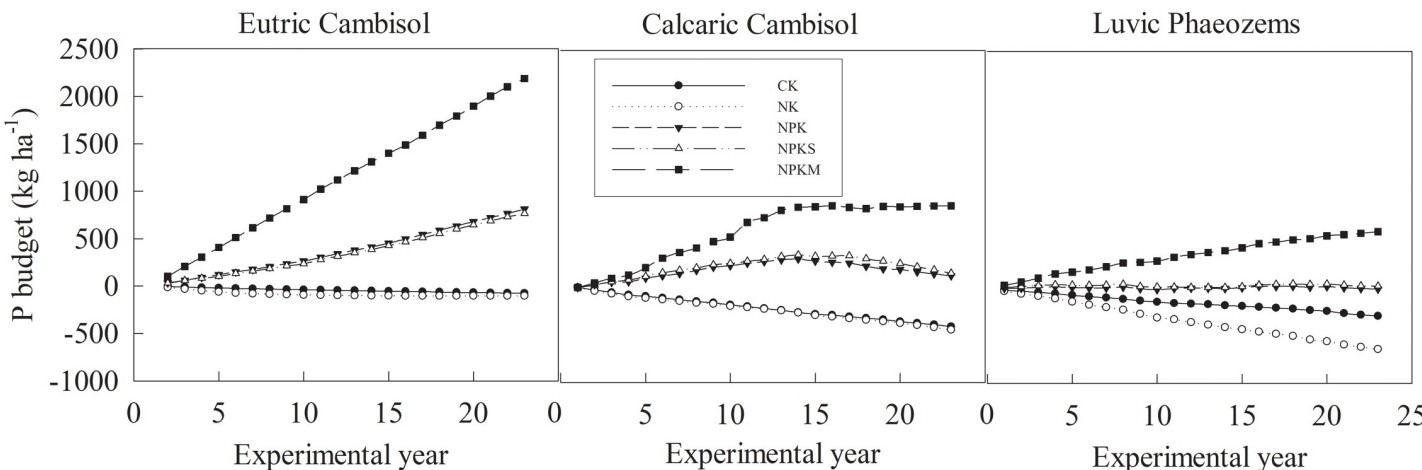

**Fig 2. P budget of soils under different fertilizer application treatments.** CK: unfertilized control; NK: nitrogen and potassium; NPK: nitrogen, phosphorous, and potassium; NPKS: nitrogen, phosphorous, potassium and straw; and NPKM: nitrogen, phosphorous, potassium and manure (NPKM).

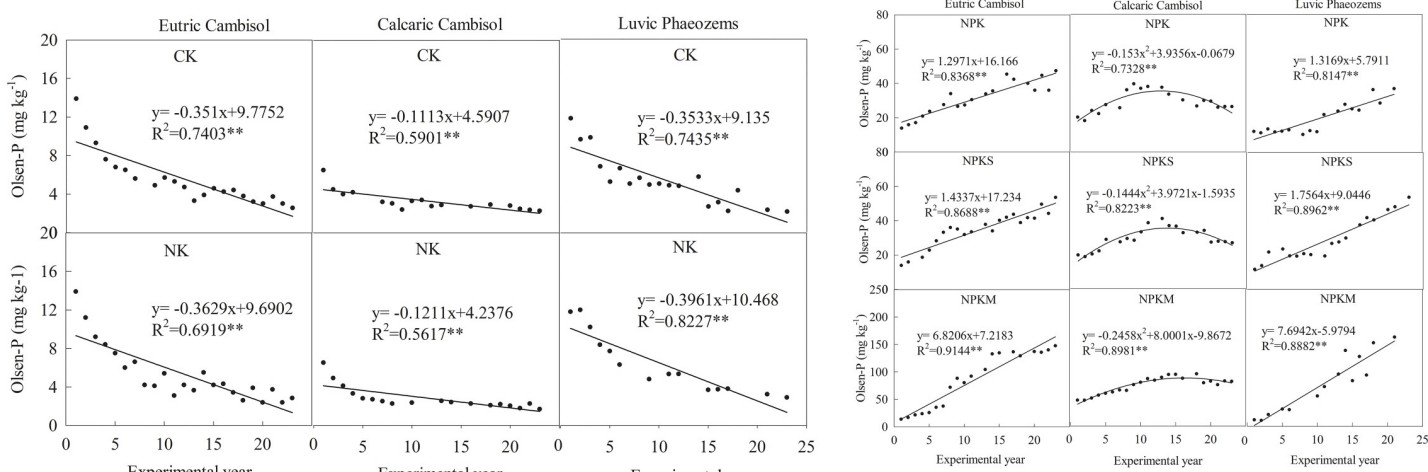

**Fig 3. Change in soil Olsen-P over time in response to different fertilization treatments.** CK: unfertilized control; NK: nitrogen and potassium; NPK: nitrogen, phosphorous, and potassium; NPKS: nitrogen, phosphorous, potassium and straw; and NPKM: nitrogen, phosphorous, potassium and manure (NPKM).

(Fig 3). Under the treatments without P applications (CK and NK), soil Olsen-P exhibited a significantly ($P<0.01$) negative correlation with the experimental years in the three sites, and the slope of the equation indicated the rates of soil Olsen-P decreasing. The order of the decreasing soil Olsen-P was Luvic Phaeozems (0.37 mg kg$^{-1}$) > Eutric Cambisol (0.36 mg kg$^{-1}$) > Calcaric Cambisol (0.12 mg kg$^{-1}$) (average value over CK and NK).

Under different P fertilizer treatments (NPK, NPKS, and NPKM), the relation between soil Olsen-P and the experimental years could be simulated by simple linear models in Eutric Cambisol and Luvic Phaeozems. The order of the soil Olsen-P increase during the experimental years was Luvic Phaeozems (3.59 mg kg$^{-1}$) > Eutric Cambisol (3.18 mg kg$^{-1}$) (average value over the three treatments). In Calcaric Cambisol, the relation between soil Olsen-P and the experimental years could be simulated by quadratic equation of one unknown, while soil Olsen-P increased during the first 14 years and then decreased for the lower P input after 14-year fertilization.

## Response of soil Olsen-P to the P budget

Soil Olsen-P was primarily affected by the P budget (Fig 4). Under the treatments without P fertilizers (CK and NK), the relationship between soil Olsen-P and the P deficit could be simulated by a simple linear model. Soil Olsen-P decreased by 0.44–9.19 mg kg$^{-1}$ for every 100 kg P ha$^{-1}$ of P deficit, and the order of the decrease rate in soil Olsen-P was Eutric Cambisol (8.06 mg kg$^{-1}$) > Luvic Phaeozems (1.97 mg kg$^{-1}$)> Calcaric Cambisol (0.47 mg kg$^{-1}$) for the three soil types. For the two treatments, the decreasing order of the soil Olsen-P rates was CK>NK in Eutric Cambisol and Luvic Phaeozems; there was no significant difference of CK and NK in Calcaric Cambisol.

Under the P fertilizer treatments (NPK, NPKS, and NPKM), soil Olsen-P was significantly positive linear related to the P surplus in Eutric Cambisol and Calcaric Cambisol. With 100 kg ha$^{-1}$ P surplus, soil Olsen-P increased in the order of NPKM (6.88 mg kg$^{-1}$) > NPKS (3.66 mg kg$^{-1}$) > NPK (3.24 mg kg$^{-1}$) in Eutric Cambisol, while the order was NPK (7.27 mg kg$^{-1}$)> NPKS (6.78 mg kg$^{-1}$) > NPKM (5.33 mg kg$^{-1}$) in Calcaric Cambisol. In Luvic Phaeozems, the relation between soil Olsen-P and the P surplus under NPK and NPKS was not obvious, and under NPKM, soil Olsen-P increased by 29.50 mg kg$^{-1}$ with 100 kg ha$^{-1}$ P surplus.

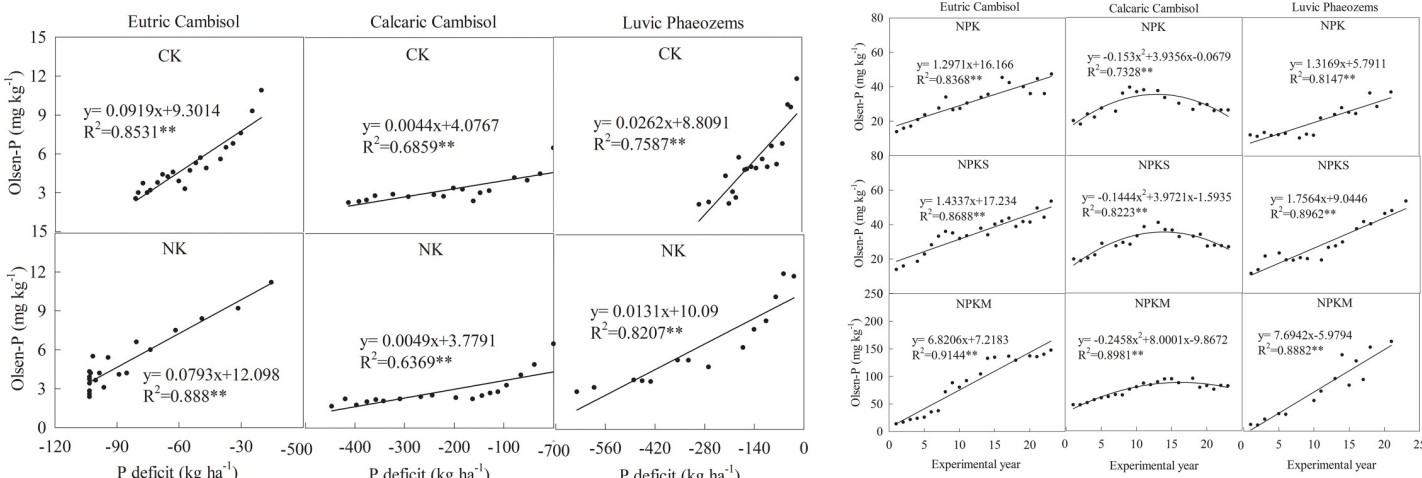

**Fig 4. Response of soil Olsen-P to the soil P budget under different long-term fertilizer application treatments.** CK: unfertilized control; NK: nitrogen and potassium; NPK: nitrogen, phosphorous, and potassium; NPKS: nitrogen, phosphorous, potassium and straw; and NPKM: nitrogen, phosphorous, potassium and manure (NPKM).

## Crop yield

The crop yield data represented the average data for every 5 years from 1990–2012 (Fig 5). In the treatments without P fertilizers, the crop yield decreased over the experimental years. In Eutric Cambisol, crop yield was significantly lower than that in Calcaric Cambisol and Luvic Phaeozems under CK and NK (Table 3). And the crop yield was close to zero after 15 years of fertilization under CK and NK in Eutric Cambisol. The crop yield under NK was significantly higher ($P<0.05$) than that under CK in Calcaric Cambisol and Luvic Phaeozems (Table 3).

Under the treatments containing P fertilizers (NPK, NPKS, and NPKM), the crop yield was improved quite a bit at the three sites. In Eutric Cambisol, the crop yield under NPKM was significantly higher ($P<0.05$) than it was under NPK and NPKS (Table 3). In addition, the crop yield decreased after 10 years under the NPK and NPKS treatments; the crop yield would increase or remain within a stable level of 5000 kg ha$^{-1}$ under NPKM in Eutric Cambisol. In Calcaric Cambisol and Luvic Phaeozems, the crop yields under the three treatments containing P fertilizers were close. In addition, in a comparison of the crop yields under NK, the crop yields did not increase a lot by a substantial amount under treatments containing P fertilizers in Luvic Phaeozems.

## Soil Organic Matter (SOM)

The soil SOM represents the average data for every 5 years from 1990–2012 (Fig 6). The soil SOM content was lowest in Calcaric Cambisol and highest in Luvic Phaeozems. Compared with the soil SOM content under treatments without P fertilizers (CK and NK) at each site, the soil SOM was higher under the treatments containing P fertilizers (NPK, NPKS, and NPKM), especially under the NPKM treatment. In Eutric Cambisol, the SOM was projected to increase during the first 15 years and then decrease after 15 years. However, the decrease was small. In Calcaric Cambisol, the SOM under CK, NK, and NPK fluctuated with the experimental years and increased under NPKS and NPKM. In Luvic Phaeozems, the SOM remained stable during the first 15 years and then increased after 15 years under all the treatments. The SOM content was significantly higher ($P<0.05$) under NPKM than it was under other P fertilizer treatments in Eutric Cambisol and Luvic Phaeozems (Table 3). At the three sites, the soil Olsen-P

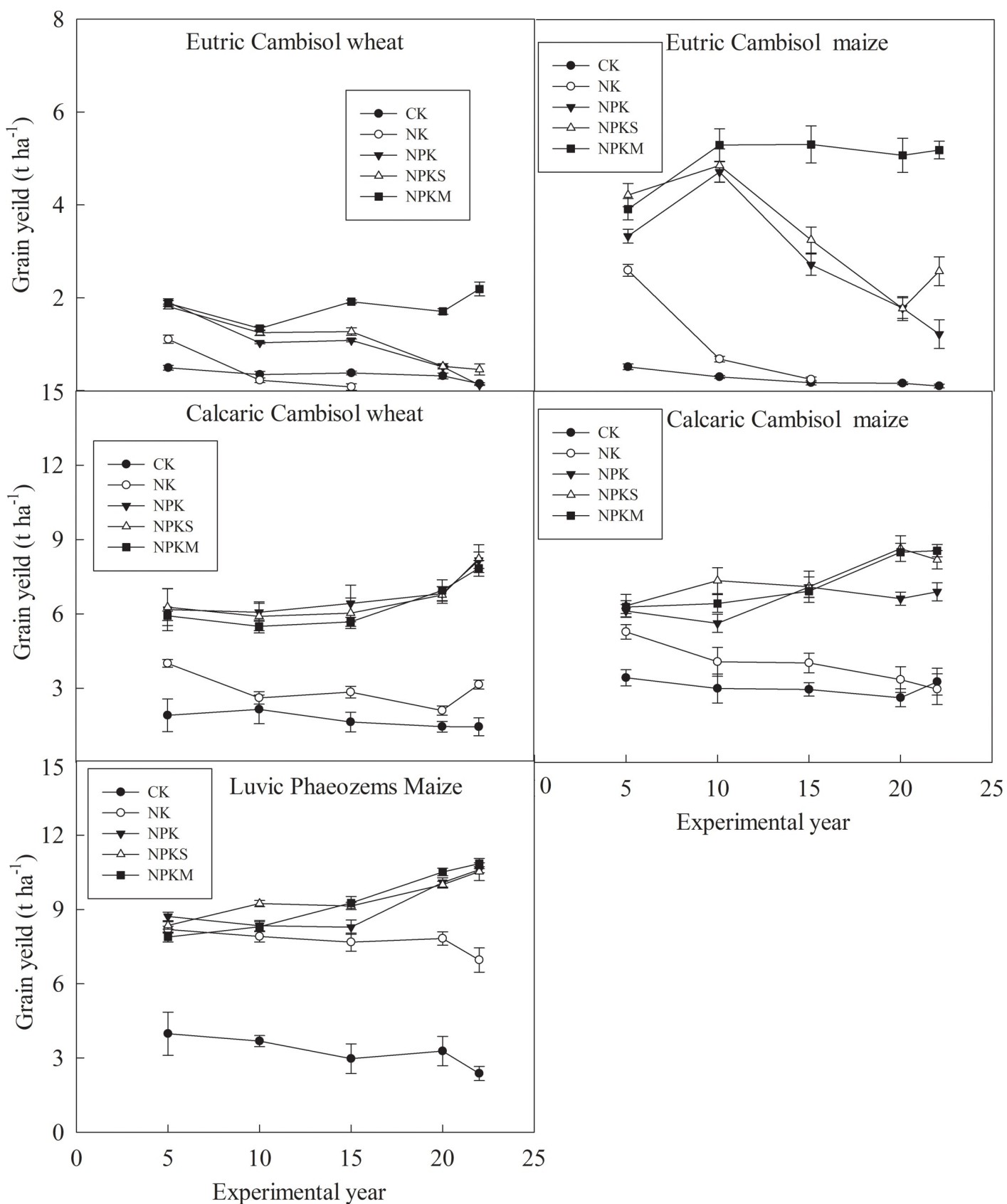

**Fig 5. Crop yields at the three sites.** CK: unfertilized control; NK: nitrogen and potassium; NPK: nitrogen, phosphorous, and potassium; NPKS: nitrogen, phosphorous, potassium and straw; and NPKM: nitrogen, phosphorous, potassium and manure (NPKM).

efficiency (the change in soil Olsen-P with a 100 kg ha$^{-1}$ P budget) was significantly ($P < 0.01$) and positively linearly related to the SOM (Fig 7).

**pH.** The soil pH represented the average data for every 5 years from 1990–2012 (Fig 8). The soil pH was lowest in Eutric Cambisol and highest in Calcaric Cambisol. The change in the soil pH was different at the three sites. In Eutric Cambisol and Luvic Phaeozems, the soil pH decreased under the NK, NPK, and NPKS treatments. However, the soil pH fluctuated around its initial value under the CK and NPKM treatments (Fig 8, Table 3). In Calcaric Cambisol, the soil pH showed the same tendency in the five treatments over a range of 8.13–8.58. The pH value would increase over the first 15 years, then decrease from 15 years to 20 years, and increase after 20 years. However, the change was small. At the three sites, the relation between the soil Olsen-P efficiency (the change in soil Olsen-P with the 100 kg ha$^{-1}$ P budget) and the soil pH was not obvious (Fig 9).

## Discussion

### Soil P budget and crop yield

Consistent with previously published studies, the P budget was calculated by taking the P input (P fertilizers) minus the P output (crop uptake). The P inputs from irrigation as well as dry and wet depositions and the P output from the P loss were neglected, as in other studies [4, 18]. The P budget is an important factor for evaluating the P management in agroecosystems, and it affects the soil P changes over time [23, 24]. At a global scale, P deficits cover 30% while P surplus covers 70% of the total cropland [3]. Long-term fertilization experiments could provide information about how the accumulated P is changing over different soil types, cropping systems and fertilization treatments.

Lack of fertilizers caused the decrease of soil fertility, for example in some parts of Tibet and Inner Mongolia, crop yields were below the national average in China [25]. Under the treatments without P fertilizers (CK and NK) in our study, the crop yield decreased over the experimental years and the soil P showed a deficit for the lack of P. Soil pH decreased to 4.0–

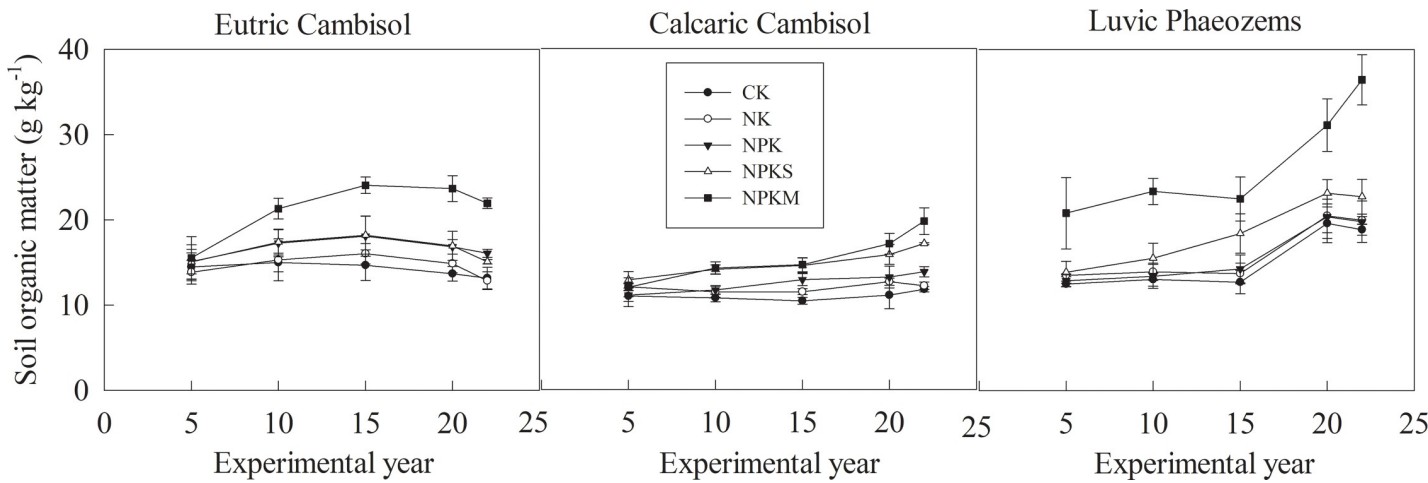

**Fig 6. Soil organic matter of the three sites.** CK: unfertilized control; NK: nitrogen and potassium; NPK: nitrogen, phosphorous, and potassium; NPKS: nitrogen, phosphorous, potassium and straw; and NPKM: nitrogen, phosphorous, potassium and manure (NPKM).

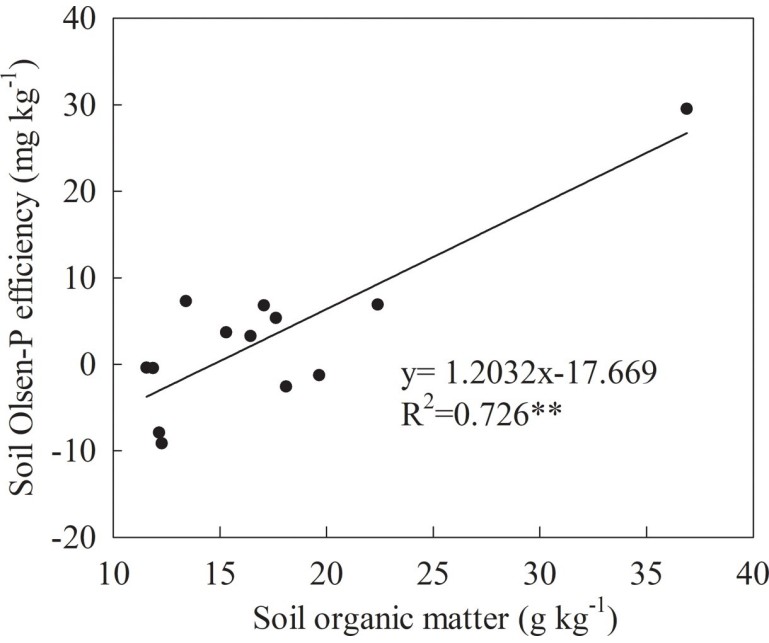

**Fig 7. The relation between soil organic matter and soil Olsen-P efficiency of the three sites.**

4.5 after 10 cropping years under CK and NK, and this caused yield reduction of (38.7–93.9%, $P < 0.05$) compared to NPK, NPKS and NPKM treatments. The addition of nitrogen caused soil acidification and it was so serious that caused a decrease of crop production, and almost eliminated yield from 2006 to 2012 under NK. Thus, the P deficit in QY was the lowest among the three sites. Lime could be applied to improve soil pH. In Luvic Phaeozems, the high initial soil Olsen-P (11.5 mg kg⁻¹) was very close to the critical P value (CPV) of maize (12.1–14.3 mg kg⁻¹) in this place [26], which could provide the crops with enough available P. Thus, the maize yield was very high in Luvic Phaeozems among the three sites. Compared to the crop yield under NK, crop yield under CK was much lower in Luvic Phaeozems, so it was thought that the lack of nitrogen was the limiting factor for crop growth in Luvic Phaeozems. When

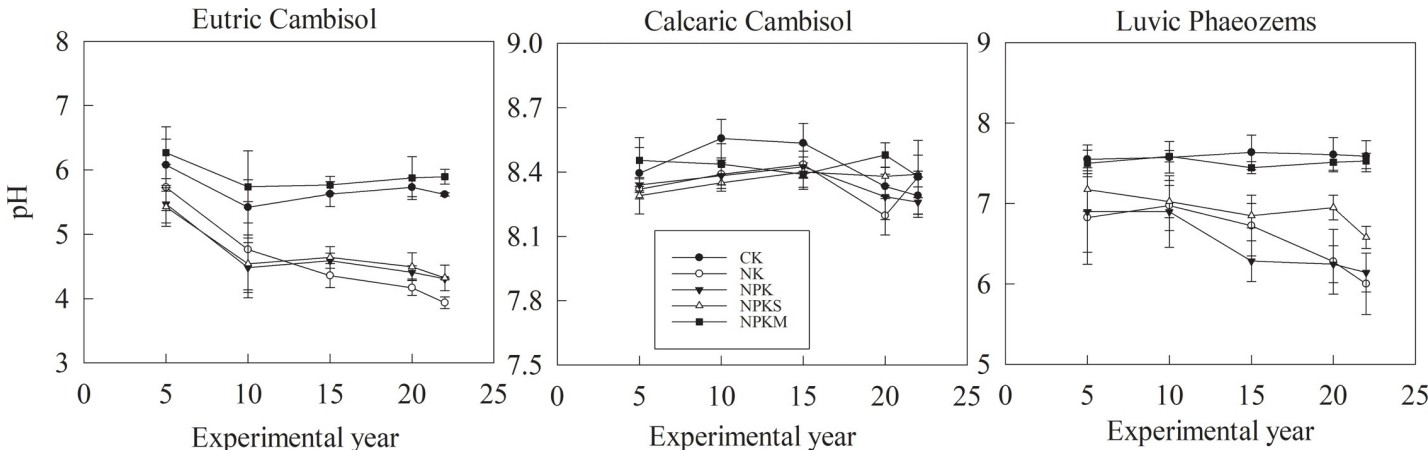

**Fig 8. CK: Unfertilized control; NK: Nitrogen and potassium; NPK: Nitrogen, phosphorous, and potassium; NPKS: Nitrogen, phosphorous, potassium and straw; and NPKM: Nitrogen, phosphorous, potassium and manure (NPKM).**

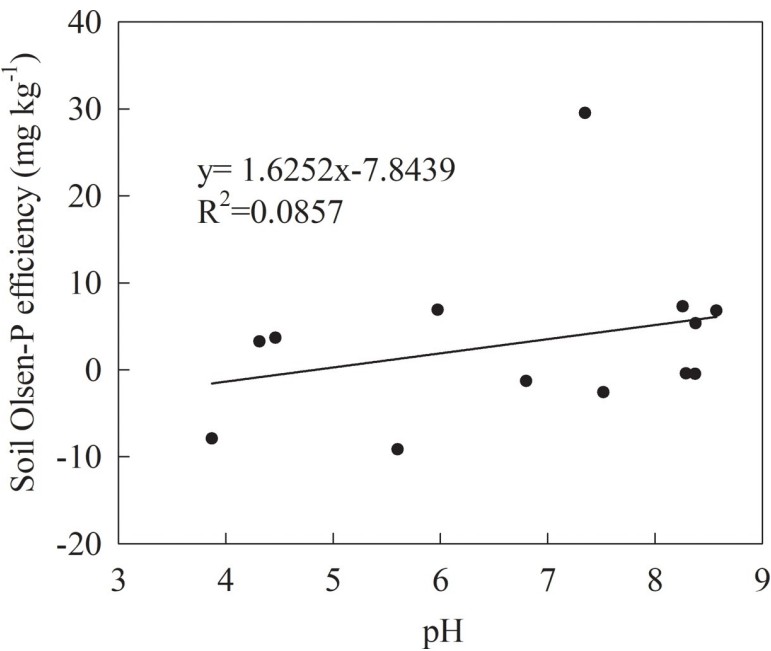

**Fig 9. The relation between soil pH and soil Olsen-P efficiency of the three sites.**

the amount of P fertilizers was higher than crop uptake, soil P showed a surplus. Although the amount of phosphorus fertilizer ($P_2O_5$) input per unit area for food crops in China has reached to 71 kg ha$^{-1}$, higher than that in many developed countries [27]. However, at least 70%-90% of the phosphorus applied to the soil was fixed in combination with Fe, Al and Ca [28]. In our study, after long-term P fertilization (NPK, NPKS, and NPKM), the soil P showed an obvious surplus in the three sites, except under the NPK and NPKS treatments in Luvic Phaeozems. The seasonal utilization rate of phosphate fertilizer in China was only 10–25% [29]. Soil P surplus was increasing by an annual growth rate of 11%. Withholding P applications, it may be feasible that accumulated soil P will build up soil P reserves for crop growth [30]. Thus, taking advantage of soil accumulated P not only can reuse and save P resource, but also can reduce the risk of environmental pollution.

Crop yield was influenced by the amount and kind of P fertilizers. In Eutric Cambisol, crop yield under NPKM was obviously higher than that under NPK and NPKS, for the input of manure alleviated soil acidification. Soil pH under NPKM was almost the same with that under CK. But P surplus was very large under NPKM in Eutric Cambisol. In Calcaric Cambisol, although P surplus under NPKM was much higher than that under NPK and NPKS, crop yields under the three treatments were almost the same. So, it was thought P acculated in Calcaric Cambisol was hard to be transformed into available P for high $CaCO_3$ content [Table 1]. In Luvic Phaeozems, soil surpluses were very low, almost to zero, but crop yields under the NPK and NPKS were almost the same with that under NPKM (NPK 8990 kg ha$^{-1}$; NPKS 9199 kg ha$^{-1}$; NPKM 9101 kg ha$^{-1}$). Low P input with high crop yield indicated that P use efficiency was very high under NPK and NPKS in Luvic Phaeozems.

## Soil Olsen-P and its relationship to the P deficit

Without P fertilizer (CK and NK), the soil Olsen-P content significantly decreased ($P < 0.01$) at the three sites. In other studies, the speed of the soil Olsen-P decreasing was decided by the

initial soil Olsen-P content, and the rate of the soil Olsen-P decrease had a positive relationship to the initial Olsen-P content [31, 32]. However, in our study, the order of the decreasing soil Olsen-P rate (from the mean of the two treatments) at the three sites was Luvic Phaeozems (0.37 mg kg$^{-1}$) > Eutric Cambisol (0.36 mg kg$^{-1}$) > Calcaric Cambisol (0.12 mg kg$^{-1}$), which was slightly different from the order of the initial soil Olsen-P (Luvic Phaeozems > Calcaric Cambisol > Eutric Cambisol). The result might be related to the amount of precipitation for Al-P, Fe-P and O-P at the soil surface was easily lost with rainfall (surface runoff) in Hunan province, especially for upland red soil [33]. The precipitation in Eutric Cambisol was two times that of Calcaric Cambisol (Table 1), so soil P loss with rainfall in Eutric Cambisol could be higher than that in Calcaric Cambisol. The speed of soil Olsen-P decreasing under NK was faster than the under CK in the three sites, for crop yield under NK was higher than that under CK. And crop took more P from soil under NK for the addition of N and K could also promote the growth of crop in the lack of P fertilizer.

Under treatments without P fertilizers, the relationship between soil Olsen-P and the P deficit could be simulated by simple linear model. With a 100 kg ha$^{-1}$ P deficit, the order of the decrease in soil Olsen-P was Eutric Cambisol (8.56 mg kg$^{-1}$) > Luvic Phaeozems (1.97 mg kg$^{-1}$) > Calcaric Cambisol (0.47 mg kg$^{-1}$). This order represents the ability of original P in the soil to convert into available P (Olsen-P), which might be affected by soil properties. The relatively lower soil pH in Eutric Cambisol could promote the dissolution of soil original P, so the rate of the soil Olsen-P decrease was largest among the three sites with the same P deficit. Compared with Calcaric Cambisol, the organic matter in Luvic Phaeozems was higher, which could promote the transformation of soil organic and inorganic P into Olsen-P [20]. Therefore, the decrease in the Olsen-P in Luvic Phaeozems was higher than that in Calcaric Cambisol under treatments without P fertilizers. Additionally, in the treatment without P fertilizers, soil P mineralization is an important pathway for replenishing the soil available P. Higher soil organic matter generally resulted in greater microbial biomass and activity, which could promote soil P mineralization [34].

## Soil Olsen-P and its relationship to the P surplus

When P fertilizers were applied (NPK, NPKS and NPKM), the soil Olsen-P content increased over the experimental years, especially in the NPKM treatment. Similar results have been reported in other articles [18, 35, 36]. In Calcaric Cambisol and Luvic Phaeozems, soil Olsen-P was positively linearly related to the experimental years, and the order of soil Olsen-P increasing was NPKM > NPKS > NPK in the two sites respectively. The result indicated that the combination of manure could improve the increase in soil Olsen-P with the 100 kg P ha$^{-1}$ surplus. However, soil Olsen-P in Calcaric Cambisol had a quadratic correlation at the 22st experimental year for the lower P input after 14 years of fertilization. The relationship between soil Olsen-P and the P surplus could be simulated by a simple linear model under all the treatments of P fertilizers at the three sites except the NPK and NPKS in Luvic Phaeozems. And the change of soil Olsen-P represented the ability of soil accumulated P transforming into Olsen-P. In Eutric Cambisol, with the same soil P surplus, soil Olsen-P increased as the order of NPKM > NPKS > NPK, which was consistent with the content of SOM order in Eutric Cambisol shown in Fig 6. A significant positive correlation (P<0.01) was also observed between the SOM and the increase in soil Olsen-P with the 100 kg P ha$^{-1}$ surplus (Fig 7). SOM is an important factor for improving the soil P availability. Because substances such as organic acid, organic anions, and humic acid would be released during the decomposition of SOM, which competed with phosphates for adsorption sites on the surfaces of soil colloids through processes such as competitive adsorption and chelation [37, 38, 39]. The P fertilizers adsorbed by

minerals could be decreased [40]. The content of SOM could also explain the order of soil Olsen-P increasing as Luvic Phaeozems > Eutric Cambisol > Calcaric Cambisol under NPKM with the same P surplus. The soil P availability was relatively high in the pH range from 6.0–7.5. The addition of manures could adjust the pH of the soil in Eutric Cambisol with a range of 5.5–6.0 [41, 42], so the increase of soil Olsen-P with same P surplus under NPKM was much higher than that under NPK and NPKS.

In Calcaric Cambisol, with the same soil P surplus, the order of soil Olsen-P increasing was NPK > NPKS > NPKM, but the value did not vary a lot. Although P surplus under NPKM was significantly higher than that of NPK and NPKS in Calcaric Cambisol, soil Olsen-P did not improve a lot for high pH. Soil pH value of the three treatments in Calcaric Cambisol was in the range of 8.25–8.50, much higher than the high P availability range of 6.0–7.5. In alkaline calcareous soils with high pH values, the inorganic phosphates (primarily as $HPO_4^{2-}$) in the soil solutions combine with Ca to form a series of Ca-P compounds [43, 44]. In Calcaric Cambisol of our study, $Ca_{10}$-P and O-P which were hard to decompose might be the main compounds of soil P. Soil containing large quantities of clay fix more P than soil with low clay contents [45, 46], the soil clay content was strongly related to increasing the Olsen-P across seven long-term experimental sites in China, except for the Huan site [4]. But in Calcaric Cambisol, the content of clay was low, so the adsorption of P might be low, precipitation could be a main way for P fixing in the soil of Calcaric Cambisol.

Under the NPK and NPKS treatments in Luvic Phaeozems, the P surplus was relatively low, but soil Olsen-P also increased. The result was the same as Zhan's [18]. The soil total P in the NPK and NPKS treatments increased to some extent (NPK: 5% and NPKS: 18%) over the 21-year experiment, which verified the result of the P budget in the two treatments. The result could be explained by the transformation of organic P (Po) into inorganic P (Pi) fractions, and the high SOM content could promote this process [33]. Under NPK and NPKS, the Po decreased by 50% and 56%, respectively, and the Pi increased by 34% and 57%, respectively [20], during the 21-year experiment. In addition, a high clay content might be another reason for the significant increase in soil Olsen-P with the low P surplus under NPK and NPKS in Luvic Phaeozems. So, in our study, the relationship between soil Olsen-P and P budget was decided by many reasons, such as the amount of P fertilizers, the kinds of fertilizers, soil properties (SOM, pH, clay content), and climate [47]. In the practice of fertilization, it's not necessary to increase the amount of P fertilizers, farmers should take measure to solve the local problem, for adjust the soil pH of Eutric Cambisol and Calcaric Cambisol. And in the area of a lot soil P surplus, it is encouraged to stop fertilization for a few years to take advantage of accumulated P and make the high Olsen-P content decrease to a reasonable level.

## Conclusions

1. Crop yield was not consistent with P budget and soil Olsen-P in the three sites for the low soil pH value in Eutric Cambisol, and high content of soil initial Olsen-P and SOM in Luvic Phaeozems. In each site, crop yield under treatments with P fertilizers was significantly higher than that under treatments without P fertilizers. In Eutric Cambisol, crop yield under NPKM was significantly higher than that under NPK and NPKS. But in Luvic Phaeozems and Calcaric Cambisol, crop yield under NPK, NPKS, and NPKM was close, which caused by higher content of $CaCO_3$ in Calcaric Cambisol, and high soil initial Olsen-P and SOM in Luvic Phaeozems.

2. Under treatments without P fertilizers, soil Olsen-P and the P deficit had a significantly positive relationship. With every 100 kg P ha$^{-1}$ of deficit, the order of the soil Olsen-P

decrease was Eutric Cambisol (8.56 mg kg$^{-1}$) > Luvic Phaeozems (1.97 mg kg$^{-1}$)> Calcaric Cambisol (0.47 mg kg$^{-1}$). The order represents the ability of original P in the soil to convert into available P (Olsen-P), which could be affected by pH and SOM.

3. Under treatments with P fertilizers, the relation between soil Olsen-P and the P surplus could be simulated by a simple linear equation except under NPK and NPKS in Luvic Phaeozems. With 100 kg ha$^{-1}$ in P surplus, soil Olsen-P increased by 3.24–7.27 mg kg$^{-1}$ in the order Calcaric Cambisol (6.42 mg kg$^{-1}$) > Eutric Cambisol (3.24 mg kg$^{-1}$). Under the NPK and NPKS treatments in Luvic Phaeozems, the P surplus was relatively low, but soil Olsen-P also increased. The result could be explained by that the high SOM content promoted the transformation of organic P (Po) into inorganic P (Pi) fractions.

4. In the practice of fertilization, it is reasonable to apply the kind and the amount fertilizers according to the local soil need. For example in Luvic Phaeozems, the initial soil Olsen-P was close to the CPV of crop, phosphorus was not serious needed in this place, and soil nitrogen was more needed, so crop yield under different P fertilizers (NPK, NPKS, and NPKM) was not significantly different.

## Supporting information

**S1 Data.**
(XLSX)

## Acknowledgments

We acknowledge all staff associated with these three long-term Monitoring Network of Soil Fertility and Fertilizer Effects at Qiyang, Zhengzhou and Gongzhuling for their valuable work.

## Author Contributions

**Conceptualization:** Shuxiang Zhang.

**Data curation:** Shuxiang Zhang, Ping Zhu, Chang Peng, Shaomin Huang, Boren Wang, Huimin Zhang.

**Formal analysis:** Weiwei Zhang, Shuxiang Zhang.

**Funding acquisition:** Shuxiang Zhang.

**Investigation:** Weiwei Zhang, Qiong Wang, Ping Zhu, Chang Peng, Shaomin Huang, Boren Wang, Huimin Zhang.

**Methodology:** Weiwei Zhang, Shuxiang Zhang.

**Resources:** Shuxiang Zhang.

**Software:** Weiwei Zhang, Qiong Wang, Qihua Wu.

**Supervision:** Shuxiang Zhang.

**Writing – original draft:** Weiwei Zhang, Qihua Wu.

**Writing – review & editing:** Weiwei Zhang, Shuxiang Zhang.

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
