## [Decision Letter · Decision Letter 0]

2 Jan 2020

PONE-D-19-33392

The response of soil Olsen-P to the P budgets of three typical cropland soil types under long-term fertilization

PLOS ONE

Dear Mrs Zhang,

Thank you for submitting your manuscript to PLOS ONE. After careful consideration, we feel that it has merit but does not fully meet PLOS ONE’s publication criteria as it currently stands. Therefore, we invite you to submit a revised version of the manuscript that addresses the points raised during the review process.

We would appreciate receiving your revised manuscript by Feb 16 2020 11:59PM. To enhance the reproducibility of your results, we recommend that if applicable you deposit your laboratory protocols in protocols.io, where a protocol can be assigned its own identifier (DOI) such that it can be cited independently in the future. For instructions see: http://journals.plos.org/plosone/s/submission-guidelines#loc-laboratory-protocols

We look forward to receiving your revised manuscript.

Kind regards,

Vassilis G. Aschonitis

Academic Editor

PLOS ONE

Journal Requirements:

3. Please include your tables as part of your main manuscript and remove the individual files. Please note that supplementary tables (should remain/ be uploaded) as separate "supporting information" files.

4. We suggest you thoroughly copyedit your manuscript for language usage, spelling, and grammar. If you do not know anyone who can help you do this, you may wish to consider employing a professional scientific editing service.  

5. Please include your tables as part of your main manuscript and remove the individual files. Please note that supplementary tables (should remain/ be uploaded) as separate "supporting information" files.

6. Please ensure that you refer to Figure 9 in your text as, if accepted, production will need this reference to link the reader to the figure.

Reviewers' comments:

Reviewer's Responses to Questions

**Comments to the Author**

1. Is the manuscript technically sound, and do the data support the conclusions?

Reviewer #1: Partly

Reviewer #2: Yes

2. Has the statistical analysis been performed appropriately and rigorously? 

Reviewer #1: No

Reviewer #2: Yes

3. Have the authors made all data underlying the findings in their manuscript fully available?

Reviewer #1: Yes

Reviewer #2: Yes

4. Is the manuscript presented in an intelligible fashion and written in standard English?

Reviewer #1: No

Reviewer #2: Yes

5. Review Comments to the Author

Reviewer #1: It is an interesting article, which I strongly recommend to be revised and submitted again, as it includes data for many years from crop yield, phosphorus fertilization and phosphorus budget. However, there are some comments I would like to make: The statistical analysis has not been performed appropriately. I would recommend that the authors conduct a Repeated measures ANOVA with three factors for Maize: Soil type (Eutric Cambisol, Calcaric Cambisol and Luvic Phaesoms), P budget and time (year as time factor) to claim the effect of long-term (time) phosphorus fertilization (phosphorus budget) over different soil types (Soil type). You can then present the results in different graphs (as you have already presented them) but put the error bars that result from the Repeated measures analysis. In the text you must explain if there are any interactions and provide the p value of the analysis. You will then conduct the same analysis for wheat only for Qiyang and Zhengzhou regions.

There are also some comments below showing some examples of revisions that have to be made both for statistics and English (the language is not very clear at some points):

Line 106: “In order to analyses in spatial variation” should be written as follows:

In order to analyze the spatial variation …

Line 117: “Soil samples from the arable layer (0–20 cm) were collected randomly from each site” should be written as follows:

The initial soil samples, before treatment application, were collected randomly from the arable layer (0-20 cm) from each site in 1990.

Line 164: Conduct a Repeated measures ANOVA (with year as time factor) in order to claim P budget was lower in Luvic Phaeozems compared to Eutric Cambisol. LSD and t-student test are not enough when you have repeated measures.

Line 259: Conduct a Repeated measures ANOVA with three factors for Maize: Soil type (Eutric Cambisol, Calcaric Cambisol and Luvic Phaesoms), P budget and time (year as time factor) to claim the effect of long-term (time) phosphorus fertilization (phosphorus budget) over different soil types (Soil type). You can then present the results in different graphs (as you have already presented them) but put the error bars that result from the Repeated measures analysis. In the text you must explain if there are any interactions and provide the p value of the analysis. You will then conduct the same analysis for wheat only for Qiyang and Zhengzhou regions.

Line 266: Rephrasing: “The variation of crop yield was affected by soil pH obviously in Eutric…” should be written as follows: Soil pH decreased to 4.0-4.5 after 10 cropping years under CK and NK, and this caused yield reduction of (provide percentage and p value) compared to NPK, NPKS and NPKM treatments.

Line 268: Rephrasing: “Soil acidification was so serious to cause a decrease of crop production, even no…” should be written as follows:

Soil acidification was so serious that caused a decrease of crop production, and almost eliminated yield from 2006 to 2012 under NK.

Line 247: Compare to instead of "Compared with"

Line 276: When the amount of P fertilizers was higher than crop uptake instead of "was higher the crop uptake".

Line 286: Thus, taking advantage instead of "Thus, taking advantaging"

Figure 2: I would recommend replacing the title of the graphs with Eutric Cambisol, Calcaric Cambisol and Luvic Phaesoms instead of Qiyang, Zhengzhou and Gongzhuling.

Figure 2: You need to provide error bars for the various treatments for each of the experimental years. You also need to conduct a repeated measures ANOVA (with year as the time factor) in order to claim the difference between the treatments.

Figure 6 and Figure 8: ** represents P<0.01. You need to remove this, because there are not error bars on the graph.

Reviewer #2: Comments:

The key finding of this manuscript is the response of soil Olsen-P to the P budgets of three typical cropland soil types under long-term fertilization. Research reported in this manuscript fits well with the general scope of the journal. The results from three relatively long period of experiments seems valuable for understanding the mechanism of the variation of soil Olsen-P affected by different fertilization practices. However, only the P in the 0-20 cm soil is considered in the P budget in this research. The reason should be provided in the revised manuscript. Besides, there are several mistakes in the current version of the manuscript. Hence, I truly believe that a major revision is needed before the manuscript is reconsidered for the publication in this journal.

Specific comments:

L28: ‘g kg-1’? Please check.

L35: ‘g kg-1’? Please check.

L36: why were only two types of soils compared?

L40: the soil Olsen-P increased by 3.24-7.24 mg kg-1. However, the value of ‘7.24’ was not found in the following sentence.

L100: the sentence is incomplete. Please rewrite it.

L138: what is the applied soil P?

L139: the P from the fertilizer was not calculated in the formula?

L145: Not all the experiments have three replications (L105).

L151 Why and how was the data were normalized? It is not clear.

L186: replace ‘mean value of the three treatments’ as ‘average value over the three treatments’

L267:replace ‘，’ as ‘,’

L267: Is the soil acidification correlated to the fertilizer application practices? What is the reason for the serious soil acidification?

L277: the sentence ‘When the amount of P fertilizers was higher the crop uptake…’ is misleading. Please rewrite it.

L278: 71 kg of phosphorus fertilizer? Or the P2O5. Please recheck. The number seems too low.

L394-408: the conclusion is too similar to the result of the experiments. It's hard to see the academic value of research in the current version. Please rewrite it.

6. PLOS authors have the option to publish the peer review history of their article (what does this mean?). If published, this will include your full peer review and any attached files.

Reviewer #1: Yes: Miltiadis Iatrou

Reviewer #2: No

---

## [Author Response · Author response to Decision Letter 0]

6 Feb 2020

Dear reviewers,

Thank you very much for your comments of our manuscript entitled “The response of soil Olsen-P to the P budgets of three typical cropland soil types under long-term fertilization”. Those comments are very valuable for revising and improving our paper. About the English grammar and writing, I have the manuscript polished by AJE and a teacher from England in CAAS help me. The manuscript is revised submission with new line and page numbers in the text, some grammar and spelling errors had also been corrected. Based on your suggestion, we made correction carefully. The responds to the comments are as follows:

According to Reviewer #1:

1. The statistical analysis has not been performed appropriately. I would recommend that the authors conduct a Repeated measures ANOVA with three factors for Maize: Soil type (Eutric Cambisol, Calcaric Cambisol and Luvic Phaesoms), P budget and time (year as time factor) to claim the effect of long-term (time) phosphorus fertilization (phosphorus budget) over different soil types (Soil type). You can then present the results in different graphs (as you have already presented them) but put the error bars that result from the Repeated measures analysis. In the text you must explain if there are any interactions and provide the p value of the analysis. You will then conduct the same analysis for wheat only for Qiyang and Zhengzhou regions.

Answer: Thank you for your helpful suggestion, about the repeat measure. There was no replication of P budget, soil Olsen-P, crop yield, SOM and pH of each year in the three sites. We only have one data of the indicators before in the long-term experiment, which I have corrected in the manuscript. We have done the ANOVA analysis for the effects of soil type, fertilization, and soil type and fertilization interactions in Table 3, the data of each year was repitation. And we have added the error bars in the Fig 5,6,8 for the average value of 5 years of crop yield, SOM, and pH. 

2. There are also some comments below showing some examples of revisions that have to be made both for statistics and English (the language is not very clear at some points):

1) Line 106: “In order to analyses in spatial variation” has been written as follows:

Line 111: In order to analyze the spatial variation …

2) Line 117: “Soil samples from the arable layer (0–20 cm) were collected randomly from each site” has been written as follows:

Line 139: The initial soil samples, before treatment application, were collected randomly from the arable layer (0-20 cm) from each site in 1990.

3) Line 266: Rephrasing: “The variation of crop yield was affected by soil pH obviously in Eutric…”has been written as follows: 

Line 340: Soil pH decreased to 4.0-4.5 after 10 cropping years under CK and NK, and this caused yield reduction of (38.7 - 93.9%, P <0.05) compared to NPK, NPKS and NPKM treatments.

4) Line 268: Rephrasing: “Soil acidification was so serious to cause a decrease of crop production, even no…” has been written as follows:

Line 342: Soil acidification was so serious that caused a decrease of crop production, and almost eliminated yield from 2006 to 2012 under NK.

5) Line 247: Compare with has been changed into Line 348"Compared to"

6) Line 351: When the amount of P fertilizers was higher than crop uptake instead of "was higher the crop uptake".

7) Line 361: Thus, taking advantage instead of "Thus, taking advantaging"

8) The titles of the Fig2,3,4,5,6,8 have been replaced by Eutric Cambisol, Calcaric Cambisol and Luvic Phaesoms (instead of Qiyang, Zhengzhou and Gongzhuling).

9) Figure 6 and Figure 8: ** represents P<0.01 has been removed.

Reviewer #2: 

1. Only the P in the 0-20 cm soil is considered in the P budget in this research. The reason should be provided in the revised manuscript. 

Answer: In the article “Storage, Patterns and Environmental Controls of Soil Phosphorus in China”, because the soil total P pool was mainly within this layer of the three sites.

2. There are several mistakes in the current version of the manuscript. 

Specific comments:

1) L28 and L35: g kg-1 has been changed into “mg kg-1”

2) L36: why were only two types of soils compared?

Answer: Because the relation between soil Olsen-P and the experimental years could be simulated using quadratic equation of one unknown in Calcaric Cambisol for the lower P input after 14 years of fertilization.

3) L41: the soil Olsen-P increased by 3.24-7.24 mg kg-1. However, the value of ‘7.24’ was not found in the following sentence.

Answer: 7.24 has been changed into 7.27

4) L103: the sentence is incomplete. Please rewrite it.???

Answer: The sentence before: The cropping systems were different at the three sites, with mono-maize cropping at GZL (late April to late September) and, wheat-maize double-cropping at ZZ (mid-October to early June for wheat and mid-June to late September for maize) and QY(early November to early May for wheat and early April between wheat strips to July for maize). Has been changed into “The cropping systems were mono-maize cropping at GZL (late April to late September), wheat-maize double-cropping at ZZ (mid-October to early June for wheat and mid-June to late September for maize) and QY(early November to early May for wheat and early April between wheat strips to July for maize)”.

5) L161: what is the applied soil P ? ; the P from the fertilizer was not calculated in the formula?

Answer: The applied soil P was “P fertilizer”

6) Line 166：The statistical analysis description has been changed:

Line 166-175: A simple linear model was used to determine the relationships between soil Olsen-P and the experimental years, soil Olsen-P and the P budget, soil Olsen-P efficiency, soil pH, and organic matter under various fertilization patterns, quadratic equation of one unknown was used to determine the relationships between soil Olsen-P and the experimental years in Calcaric Cambisolin Excel 2013 (Microsoft Corp, Redmond, Washington, USA). The SPSS 20.0 (International Business Machines Corporation, Armonk, NewYork, USA) was used to do the data analysis by calculating the mean, the standard error of mean of crop yield, soil pH and SOM. Before doing the ANOVA significance levels in table 3, descriptive statistics was used in SPSS to ensure the normal distribution of data.

7) L218: replace ‘mean value of the three treatments’ as ‘average value over the three treatments’

8) L342: Is the soil acidification correlated to the fertilizer application practices? What is the reason for the serious soil acidification?

Answer: The addition of nitrogen caused soil acidification and it was so serious that caused a decrease of crop production, and almost eliminated yield from 2006 to 2012 under NK.

9) L351: the sentence ‘When the amount of P fertilizers was higher the crop uptake…’ was corrected into “When the amount of P fertilizers was higher than crop uptake”

10) L352: 71 kg of phosphorus fertilizer? Or the P2O5. Please recheck. The number seems too low.

Answer: The sentence has been changed into “Although the amount of phosphorus fertilizer (P2O5) input per unit area for food crops in China has reached to 71 kg ha-1, higher than that in many developed countries [26]”. 71 kg ha-1 P2O5 per unit area for food crops was common in fertilization practice, and in the long-term experiment, the amount of P2O5 input per unit area for food crops was usually 75 kg ha-1 P2O5.

11) L470-497: the conclusion is too similar to the result of the experiments. It's hard to see the academic value of research in the current version. Please rewrite it.

Answer: the conclusion has been rewritten as:

a) Crop yield was not consistent with P budget and soil Olsen-P in the three sites for the low soil pH value in Eutric Cambisol, and high content of soil initial Olsen-P and SOM in Luvic Phaeozems. In each site, crop yield under treatments with P fertilizers was significantly higher than that under treatments without P fertilizers. In Eutric Cambisol, crop yield under NPKM was significantly higher than that under NPK and NPKS. But in Luvic Phaeozems and Calcaric Cambisol, crop yield under NPK, NPKS, and NPKM was close, which caused by higher content of CaCO3 in Calcaric Cambisol,and high soil initial Olsen-P and SOM in Luvic Phaeozems.

b) Under treatments without P fertilizers, soil Olsen-P and the P deficit had a significantly positive relationship. With every 100 kg P ha–1 of deficit, the order of the soil Olsen-P decrease was Eutric Cambisol (8.56 mg kg-1) > Luvic Phaeozems (1.97 mg kg-1)> Calcaric Cambisol (0.47 mg kg-1). The order represents the ability of original P in the soil to convert into available P (Olsen-P), which could be affected by pH and SOM.

c) Under treatments with P fertilizers, the relation between soil Olsen-P and the P surplus could be simulated by a simple linear equation except under NPK and NPKS in Luvic Phaeozems. With 100 kg ha-1 in P surplus, soil Olsen-P increased by 3.24-7.27 mg kg-1 in the order Calcaric Cambisol (6.42 mg kg-1) > Eutric Cambisol (3.24 mg kg-1). Under the NPK and NPKS treatments in Luvic Phaeozems, the P surplus was relatively low, but soil Olsen-P also increased. The result could be explained by that the high SOM content promoted the transformation of organic P (Po) into inorganic P (Pi) fractions.

d) In the practice of fertilization, it is reasonable to apply the kind and the amount fertilizers according to the local soil need. For example in Luvic Phaeozems, the initial soil Olsen-P was close to the CPV of crop, phosphorus was not serious needed in this place, and soil nitrogen was more needed, so crop yield under different P fertilizers (NPK, NPKS, and NPKM) was not significantly different.

Once again, many thanks for your generous help with our manuscript. 

Your’s sincerely, 

Shuxiang Zhang

---

## [Decision Letter · Decision Letter 1]

25 Feb 2020

The response of soil Olsen-P to the P budgets of three typical cropland soil types under long-term fertilization

PONE-D-19-33392R1

Dear Dr. Zhang,

We are pleased to inform you that your manuscript has been judged scientifically suitable for publication and will be formally accepted for publication once it complies with all outstanding technical requirements.

With kind regards,

Vassilis G. Aschonitis

Academic Editor

PLOS ONE

Additional Editor Comments (optional):

Reviewers' comments:

Reviewer's Responses to Questions

**Comments to the Author**

1. If the authors have adequately addressed your comments raised in a previous round of review and you feel that this manuscript is now acceptable for publication, you may indicate that here to bypass the “Comments to the Author” section, enter your conflict of interest statement in the “Confidential to Editor” section, and submit your "Accept" recommendation.

Reviewer #1: All comments have been addressed

Reviewer #2: All comments have been addressed

2. Is the manuscript technically sound, and do the data support the conclusions?

Reviewer #1: Yes

Reviewer #2: Yes

3. Has the statistical analysis been performed appropriately and rigorously? 

Reviewer #1: Yes

Reviewer #2: Yes

4. Have the authors made all data underlying the findings in their manuscript fully available?

Reviewer #1: Yes

Reviewer #2: Yes

5. Is the manuscript presented in an intelligible fashion and written in standard English?

Reviewer #1: Yes

Reviewer #2: Yes

6. Review Comments to the Author

Reviewer #1: I have seen that they have included the changes that I suggested and I think this publication is now ready to be submitted. I am particularly content that they reviewed their statistical analysis and incorporated changes in their text and the graphs. I can also see that they have improved a lot their English and the document looks now appropriate for publication. Anyway, this publication describes an experiment which has run for long time and clarifies some issues about phosphorus absorption and budget in soil.

Reviewer #2: (No Response)

7. PLOS authors have the option to publish the peer review history of their article (what does this mean?). If published, this will include your full peer review and any attached files.

Reviewer #1: Yes: Miltiadis Iatrou

Reviewer #2: Yes: Zhen Wang

---

## [Editor Report · Acceptance letter]

3 Mar 2020

PONE-D-19-33392R1 

The response of soil Olsen-P to the P budgets of three typical cropland soil types under long-term fertilization 

Dear Dr. Zhang:

I am pleased to inform you that your manuscript has been deemed suitable for publication in PLOS ONE. Congratulations! Your manuscript is now with our production department. 

With kind regards,

on behalf of

Dr. Vassilis G. Aschonitis 

Academic Editor

PLOS ONE